# Association of Health Status and Health Behaviors with Weight Satisfaction vs. Body Image Concern: Analysis of 5888 Undergraduates in Egypt, Palestine, and Finland

**DOI:** 10.3390/nu11122860

**Published:** 2019-11-21

**Authors:** Walid El Ansari, Gabriele Berg-Beckhoff

**Affiliations:** 1Department of Surgery, Hamad General Hospital, Doha 3050, Qatar; 2College of Medicine, Qatar University, Doha 3050, Qatar; 3School of Health and Education, University of Skövde, 541 45 Skövde, Sweden; 4Faculty of Applied Sciences, University of Gloucestershire, Gloucester GL53 7TH, UK; 5Unit for Health Promotion Research, Institute of Public Health, University of Southern Denmark, Niels Bohrs Vej 9, 6700 Esbjerg, Denmark; gbergbeckhoff@health.sdu.dk

**Keywords:** healthy nutrition, body image, health behaviors, weight satisfaction, college students, depressive symptoms, physical activity, health awareness

## Abstract

Little is known about the relationships between weight satisfaction, body image concern, healthy nutrition, health awareness, and physical activity among college students across culturally different countries. We assessed country and sex-specific associations between health status (self-rated health, depression, BMI), healthy behavior (healthy nutrition, physical activity, health awareness), weight satisfaction, and body image concern via a cross-sectional survey (5888 undergraduates) in Egypt, Palestine, and Finland. This health and wellbeing survey employed identical self-administered paper questionnaires administered at several Universities in two Eastern Mediterranean countries (Egypt, Palestine—Gaza Strip), and an online-survey comprising the same questions in Finland. Regression analyses were employed. Health status variables exhibited the strongest associations; high BMI and more depressive symptoms were more often among students satisfied with their weight (except in Palestine), but they were positively associated with body image concern irrespective of country or gender. Self-rated health was not associated with body image concern or weight satisfaction. Healthy behaviors were not associated with body image concern or weight satisfaction. Depressive symptoms and BMI were the most prominent predictors for body image concern. There were country-specific consistent results when using the body image concern score. Further research is necessary to compare body image across different cultures and countries.

## 1. Introduction

Body image, the subjective satisfaction with one’s body [1], is a multidimensional and dynamic concept where perceptions of body images and ideas are formed based on experiences, concepts, and behaviors [2]. Perceptions of poor body image could impact on an individual’s psychological wellbeing, and thus, be associated with obesity [3]. Distortion of and dissatisfaction with body image may form a link between overweight and obesity, as excessive preoccupation with appearance and pursuit of the ideal lean body can produce negative feelings and devaluation, resulting in changes in eating behavior, leading to overweight [4,5]. Body image concerns can also negatively affect social interaction, job opportunities, productivity, socioeconomic status, and psychosocial performance [6]. Body dissatisfaction is consistently rated as being among the foremost concern for young people [7].

### 1.1. Interrelations between Healthy Behaviors, Health Status, and Body Image

The complex interrelations between healthy behaviors, health status, and body image are of great interest in order to develop and monitor the effectiveness of health behavior interventions. For instance, it is recognized that nutrition behavior may be associated with body image perception [8,9,10]. Likewise, physical activity (PA) is an affirmative element in self-perception, where regular PA could have a positive body image influence [11,12]; and conversely, the extent of body image concern may motivate adherence to PA [13]. Nevertheless, extensive exercise might have negative influence: it has been associated with body dissatisfaction specifically amongst women with eating pathology [14,15]; PA often resulted in increased focus and awareness concerning weight and shape [16], and perceived sociocultural pressure and body dissatisfaction are linked with a persuasive need to exercise [17]. As for self-rated health, poor self-rated health was positively associated with body dissatisfaction and negatively associated with exercise [18,19].

### 1.2. Additional Important Variables

Among young adults, a range of other potential variables play a role in body image perception, for example, gender, religiosity, economic features, and cultural aspects. For example, body image perception and its correlates are designed and constructed differently between genders [20], where males favor muscular bodies, women desire to be lean or thin, exhibiting greater dissatisfaction when they perceive their body as bigger than desired [21]. As for religiosity, strong and internalized religious beliefs and a satisfying relationship with God were associated with lower levels of disordered eating and body image concern [22]. Similarly, cultural factors may influence self-perception [23] differentially across countries, as distinctive community or cultural aspects impact body weight norms, weight control, obesity, and body image dissatisfaction [24,25,26].

### 1.3. Knowledge Gap and the Current Study

Despite all of the above, few studies have investigated and compared body image across different countries, and cross-cultural comparison between Western and non-Western countries are needed [27]. In general, only a few of the cross-cultural studies have investigated body image and eating behavior by using the same methodology [27,28]. Research has voiced that there are no studies examining the relationships between body disaffection and PA in non- Anglo-Saxon large samples, a fact that limits the generalizability of previous findings to other cultures [29].

In terms of the cultures of different countries, beliefs about the ideal body size are acquired at an early age and proceed to develop with input from the sociocultural context [10,11,30,31]; hence, beauty standards to which both men and women compare themselves are entrenched in a society and its culture [32]. For instance, Egypt and Palestine are Mediterranean, Muslim oriented, geographically-neighboring countries with some social resemblances, although their dietary, lifestyles and attitude, and perception of obesity might vary. Conversely, there are cultural differences between young adults living within different social systems and dissimilar social changes, for example, Finland as a Nordic country with different social behavior, religiosity, social norms, and living conditions. Such views agree with postulations that self-perception of one’s body is associated with culture and social pressure [32,33], as the creation of beauty ideals happens during socialization where parents and caregivers communicate to their kids their culture-specific patterns of beauty [3,34]. Despite the fact that globalization affects the homogeneity of cultures, the native culture of each region keeps some of its trends that are maintained by internal mechanisms [33].

There is a range of variables that are important to consider when examining body image across different countries. These include, for instance, gender, BMI, body image dissatisfaction, lifestyle characteristics, such as physical activity and food environments, and mental wellbeing variables, for example, depressive symptoms, finance-related stress, as well as health awareness and self-rated health [32,33,35,36,37,38,39].

The university environment is important for studies of body image and related variables due to physiological and psychological factors that include identity and role changes, insufficient exercise, cafeteria food, and the availability and ease of snacking on junk food [40,41]. To the best of our knowledge, no comparative research examined such differences among young adult university students across these three culturally different countries, particularly involving both genders. Investigating and comparing body image concerns, BMI, PA, and nutritional habits and their interrelationships between such three countries could help to uncover unique characteristics of young people in each country that interplay to shape their body image concerns. Such information is critical in order to tailor appropriate education and intervention programs. Therefore, across undergraduate students in Egypt, Palestine, and Finland, the specific objectives of the current study were to assess:(1)Weight satisfaction and body image concern by BMI category.(2)Health status, healthy behavior, and sociodemographic predictors of weight satisfaction stratified by country and sex.(3)Health status, healthy behavior, and sociodemographic predictors of body image concern stratified by country and sex.

Little is known about the relationships between weight satisfaction, body image concern, healthy nutrition, health awareness, and PA among college students in these three countries. Understanding body image preferences and its intricate relationships are essential to molding public health obesity and healthy nutrition interventions.

## 2. Materials and Methods

### 2.1. Study Design, Ethics, and Procedures

The Research Ethics Committee at each participating institution across the three countries provided ethical approval for the study. The current student health and wellbeing survey (cross-sectional study) employed identical self-administered paper questionnaires administered at several Universities in two EMR countries (Egypt, Palestine—Gaza Strip) and an online-survey comprising the same questions in Finland [41,42,43]. Participation was anonymous, students consented for participation in the study, and all data were confidential and protected. Local organizers in the Egypt and Palestine institutions distributed the self-administered questionnaire to students of random classes at each faculty or institution; data were collected usually at the end of lectures where questionnaires were distributed to participants attending randomly selected classes and then collected after completion. Participation in the survey was completely voluntary, and to encourage participation and improve the response rates (without coercion), we highlighted to students the importance of their participation. Information sheets detailing the aim and objectives of the study accompanied each questionnaire, and students were informed that by completing the questionnaire, they were consenting to participate in the study.

In Egypt, data were collected (2009–2010) from all faculties at the University of Assiut in Assiut city (Business, Engineering, Education, Arts, Social Work, Sciences, Physical Education, Computers and Information, Veterinary Medicine, Specific Education, Agriculture). This yielded 3271 students, 659 participants were excluded due to missing data on the variables under examination. The response rate was ≈90%.

In Palestine (Gaza strip), students at 5 universities and 3 colleges participated in the study during 2013. Private, public, and governmental universities were included representing many disciplines (Al-Azhar University, Al-Aqsa University, Palestine University, The Islamic University, Gaza University, College of Applied Sciences, Palestine College of Nursing, Dar Al Dawa, Humanities College). A total of 1428 students completed the survey, 209 were excluded due to missing values on the variables under examination. The response rate was ≈85%.

In Finland, data were collected at the University of Turku, during the academic year 2013–2014. Data collection was done via universal sampling. The invited students participated by completing an online survey. The total number of completed questionnaires was 1189 (response rate ≈27%), but due to missing values in one of the variables of interest, 97 questionnaires were excluded from the current analysis.

### 2.2. Research Instruments

The study included self-reported sociodemographic information and nutritional habits (consumption frequency of 12 food groups).

#### 2.2.1. Body Weight Satisfaction and Body Image Concern

Weight satisfaction (1 item) was measured with the question, “How satisfied are you with your current weight in general?” The four initial response categories were categorized for the current analysis into ‘very or somewhat satisfied’ and ‘somewhat or very dissatisfied’ [44].

The body image concern score (8 items) was assessed based on summing up the responses of the 8 items of the body shape questionnaire [45,46], measuring levels of concern with shape in the last 4 weeks (six-point Likert scale, 1 = “never” to 6 = “always”. The items covered symptoms that could appear regarding a negative body image perception. Sample items include: “Have you been so worried about your shape that you have been feeling your ought to diet?” or “Have you noticed the shape of others and felt that your own shape compared unfavorable?” Country specific Cronbach’s alpha indicated very good to excellent internal consistencies (Table 1).

#### 2.2.2. Depressive Symptoms (19 Items)

These were measured using a Modified Beck’s Depression Inventory (M-BDI) [47,48]. Nineteen statements containing negative attitudes to live (e.g., I feel sad, or I feel annoyed and irritated) were answered using a 6-point Likert-scale ranging from 1 (never) to 6 (almost always). One question (I lost interest in sex) was not used in Egypt and Palestine, as it was culturally inappropriate. The remaining 19 variables revealed a very consistent latent variable with Cronbach’s alpha of 0.87 (Egypt), 0.90 (Palestine), and 0.94 (Finland). We defined clinically relevant depressive symptoms as the >85% percentile out of the whole sample, in line with others [48], which for the current study was equivalent to an M-BDI-score of >77.

#### 2.2.3. Healthy Nutrition Guideline Achievement (12 Items)

Using students’ FFQ responses [40], we computed an objective dietary guideline achievement index, with a maximum of 8 points (8 guidelines), derived from 8 foods: (1) sweets, cookies, and snacks; (2) fast food or canned food; (3) lemonade or soft drinks; (4) fruits; (5) salad and raw vegetables; (6) cooked vegetables; (7) meat; and, (8) fish. For sweets, cake or cookies, snacks, fast food or canned food, and lemonade or soft drinks, the description to eat such food only occasionally was interpreted as ‘1–4 times a month’ and ‘never’ as recommended. To consider all sweets, cake or cookies, and snacks together, we summed up all three categories into ‘sweets, cookies, and snacks score’, and healthy nutrition was considered present if this score was ≤6 corresponding to 3 times intake of these items of ‘less often than 1–4 times a month’. Each of the fast food or canned food and lemonade or soft drinks were included as individual items in computing the guideline achievement index. For the remaining food groups, we used the WHO dietary guidelines recommendations for the Eastern Mediterranean region [49]. Hence, for the number of daily fruit and raw and cooked vegetable servings, the cutoff was ‘daily’ or ‘several times a day’; for meat, the cutoff was ‘less than daily’; and for fish, the cutoff was ‘several times/week’. Milk and cereals were not included in computing the achievement index as milk consumption is healthy only if lactose intolerance is not present, a condition that appears quite often (74%) among Egyptian children [50]; and the information about cereals is generally too unspecific to categorize it as healthy or unhealthy nutrition.

#### 2.2.4. Physical Activity (2 Items)

Vigorous PA was measured by: “On how many of the past 7 days did you participate in vigorous exercise for at least 20 minutes?” Participants answered with 0–7 days. We used a cutoff of ≥3 days/week as an achievement of the PA guidelines [36]. Moderate PA was measured by: (“On how many of the past 7 days did you participate in moderate exercise for at least 30 minutes?” Participants answered with 0–7 days. We used a cutoff of ≥5 days/week as an achievement of the PA guidelines for adults from the American College of Sports Medicine and the American Heart Association [51].

#### 2.2.5. Health Status Variables

Health awareness (1 item): “To what extent do you keep an eye on your health?”, with four response categories, where for the current analysis, the two response categories ‘to some extent’ and ‘very much’ were grouped into “yes”, and ‘not at all’, and ‘not much’ were grouped into “no” [52].

Self-rated health (1 item): “How would you describe your general health?”, where the initial response categories ‘excellent’, ‘very good’, and ‘good’ were grouped together and compared to the categories ‘fair’ and ‘poor’ grouped together [53].

#### 2.2.6. Sociodemographic Variables

These included age, gender, respondent’s subjective economic situation (“How sufficient is your income?”, with five response options, recoded into ‘sufficient’ vs. ‘not sufficient’), religiosity (importance of religion, “How strongly do you agree with the following statement: My religion is very important to my life?”, where the students chose ‘strongly agree’ out of a total 5 response options), and body mass index (BMI), calculated from measured weight and height (kg/m^2^) in Egypt and reported weight and height (kg/m^2^) in Palestine and Finland.

### 2.3. Statistical Analysis

The analysis was conducted in SAS Version 9.4 (SAS Institute Inc., Cary, NC, USA). For descriptive purposes, we present frequency and percentages. Simple cross-tabulation assessed the association between body image concern and BMI. A multivariable logistic regression was undertaken to assess the association between weight satisfaction and healthy behaviors (moderate PA, vigorous PA, healthy nutrition, health awareness), health status (general self-rated health, depression, BMI), and relevant sociodemographic characteristics (age, economic situation, religiosity). Additionally, a multivariable linear regression analysis was conducted to assess the association between the body image concern score and the same healthy behaviors, health status, and sociodemographic variables. Model assumptions were tested graphically, and the log transformation of the outcome variable of the body image concern score was deemed necessary. We found sex and country interaction in all regression analyses, and therefore stratified analyses were undertaken. Multiple testing was considered using Bonferroni-Correction. The *p*-value was set to 0.001.

## 3. Results

### 3.1. Characteristics of the Sample by Country

Of the 5888 students who participated in this survey, 4923 were included in the current analysis. Table 1 shows that there were a larger number of participants from Egypt, and across the three countries, there were more female students, more students aged 19–21 years, and students who were generally satisfied with their economic situation (disposable monthly income). Nearly all the Arab respondents reported religiosity as very important, while in Finland, 40% of the students reported such sentiments. Whilst >60% of students were very or somewhat satisfied with their weight in Palestine, only ≈35% in both Egypt and Finland expressed such satisfaction. Body image concern (mean of log-transformed body image concern levels) did not differ across the three countries.

### 3.2. Weight Satisfaction and Body Image Concern by BMI Category

Table 2 depicts students’ BMI categories by their weight satisfaction. Unsatisfied students were overrepresented in the below normal BMI category (≤18 kg/m^2^) in Egypt but not in Palestine and Finland. There were no differences in body image concerns across the three countries. Students reporting ‘below normal’ or ‘normal’ body weight had a ≈2.6 mean log-transformed body image score, whilst it was ≈2.9 among those reporting to be ‘too overweight’.

### 3.3. Health Status, Healthy Behaviour, and Sociodemographic Predictors of Weight Satisfaction, Stratified by Country and Sex

Due to the significant interactions between gender and country and some of the variables under study, Table 3 depicts sex and country-specific stratified logistic regression analyses. The results exhibited differences by country and by sex. Self-rated health was not significantly associated with weight satisfaction. Depressive symptoms were associated with weight satisfaction only in Egypt and Finland. Moreover, high BMI was associated with being satisfied with one’s weight among Finnish students and Egyptian males, but the association was the opposite among Palestinian students. None of the healthy behaviors or sociodemographic characteristics showed a consistent or significant association with weight satisfaction. Finally, the logistic regression model’s fit was relatively low in Egypt and Palestine (max rescaled R-Square 8–20%). In Finland, the binary coded weight satisfaction was better explained (max rescaled R-Square 28% for males, 37% for females). However, the only significant predictors of weight satisfaction among Finnish students were depressive symptoms and BMI.

### 3.4. Health Status, Healthy Behaviour, and Sociodemographic Predictors of Body Image Concern Stratified by Country and Sex

Table 4 presents the health status, healthy behavior, and sociodemographic predictors of body image concern. All countries revealed similar results: depression and BMI were positively associated with worse body image concerns. The explained variance ranged between 0.20–0.35. When the MDBI score was included in the model, no consistent results in the associations between healthy behavior and body image and sociodemographic characteristics and body image were observed.

## 4. Discussion

Our two most important predictors of either weight satisfaction or body image concern were BMI and depressive symptoms. Self-rated health and health behavior, for example, PA and healthy nutrition as well as health awareness and sociodemographic characteristics, were not associated with weight satisfaction or body image concern when stratified for gender and the three participating countries.

There was no association between self-rated health and body image concern after adjustment for depressive symptoms. These results suggest that depression was a more prone predictor for body image concern than self-rated health and that depressive symptoms mediated our observed associations between self-rated health and body image concern. Self-rated health, adjusted in the multiple regression models for depressive symptoms, did not predict body image concern, in support of a cross-sectional survey in school-aged children in Australia [54]. Conversely, good self-rated health was more often seen in Finnish students unsatisfied with their weight (but not in Egypt and Palestine). Similar controversial results between self-rated health and weight satisfaction, as well as body image concern, were reported in a Korean cross-sectional study of college students [55]. Potential explanations might be that weight satisfaction is usually based on a single question and can be derived not only by feeling too overweight but also that people might be unsatisfied with being too thin, having too much or too little muscle, or having an undesirable body-shape. For our Finnish students, most of the normal weight students were unsatisfied with their weight, supporting a proposition that other factors (besides overweight dissatisfaction) might be connected with weight satisfaction. On the other hand, body image concern (measured via the body shape questionnaire) [45] is far more specifically connected to the feeling of being overweight. However, we observed a pronounced positive association between increasing BMI and worse body image concern in all countries.

Independent of BMI, depressive symptoms were positively associated with body image concern across both genders in the three countries, and negatively associated weight satisfaction across both genders in Egypt and Finland. This agreed with a population survey in Switzerland where body weight dissatisfaction was associated with depression across the overall group, as well as in men and women, independent of BMI [56]. Likewise, among Portuguese adolescents [57], body dissatisfaction contributed to depressive symptoms, without gender differences, and in the U.S. (2139 adolescent males), boys with body weight distortions reported significantly more depressive symptoms than boys without body weight distortions [58]. The links between body dissatisfaction and mental health are complex, with possible bidirectional associations. Postulated mechanisms include that body dissatisfaction stalks from an inappropriate highlighting of the importance of thinness and other unachievable beauty standards and, hence, may influence the onset of depression [59,60]. On the other hand, body image-depression links are supported by neurobiological investigations, where hypothalamic pituitary-adrenal axis and serotonin system deficits are involved in mood disorders and in weight regulation, and brain areas involved in hedonic regulation may play a role for both body image and depression [60,61].

Higher BMI was associated with more body image concern and less weight satisfaction across both genders (except for Egyptian males and Palestinian students), confirming other findings that have highlighted that BMI remains an important factor in initiatives that encourage positive body-related perceptions among adolescent girls [62]. We are also in support of university students in Italy, where higher BMI was significantly associated with a more overweight actual self-perception and increased body dissatisfaction, and BMI was a major correlate of body image perception and dissatisfaction [63]. Likewise, BMI had a significant relationship with body dissatisfaction among Iranian youth and adults [64].

In our regression analysis, health behavior (e.g., achieving healthy nutrition recommendations or achieving PA guidelines) was not associated with weight satisfaction and body image concern. For instance, moderate and vigorous PA, healthy nutrition, and health awareness were not associated with weight satisfaction or with body image concern in all three countries when we adjusted the model for other healthy behaviors, health status, and sociodemographic variables. These findings may be explained by the confounders, for example, BMI and depressive symptoms. Both variables are important confounders in the association between health behavior and body image concern and weight satisfaction, while PA, healthy nutrition, and health awareness had inconsistent direct associations with body image concern, which we observed in bivariate analyses (data not presented), before adjustment for the health status variables. Such findings resonate with the USA, where moderate-to-vigorous PA and vigorous PA were negatively correlated with body image discrepancy among 10–15 years old schoolgirls, but correlations were not significant after adjusting for race and BMI [65]. Likewise, among Canadian adolescents, BMI was a significant moderator of the association between body weight dissatisfaction and adherence to PA recommendation [66]. Whilst the underlying causes for such suppression effects necessitate further examination, others, nevertheless, found that body image dissatisfaction was significantly associated with higher moderate-to-vigorous PA participation among females aged 18–23 years [67].

We observed differences across the three countries, where the most pronounced country differences were in terms of weight satisfaction but not in terms of body image score. Hence, we agree with other reports that highlight culture as a very important factor that differentiates attitudes towards bodies [32]. Furthermore, surprising findings were observed regarding self-rated health among Finnish students, even though the results were not significant (good self-rated health was more often among those unsatisfied with their weight). Other surprising results were related to depressive symptoms and BMI in Egypt and Finland (more depressive symptoms and higher BMI were associated with being satisfied with their weight), which was not present among Palestinian students. Furthermore, BMI was negatively associated with weight satisfaction in Egypt and Finland but positively associated among Palestinian students. It is difficult to speculate whether and why there are different body image ideals in different countries and their links to prevalent sociocultural aspects or religion. In many low-to-middle-income countries (of which Egypt and Palestine are examples), stoutness has traditionally implied good health [68], although as modernization increases globally, ideologies and perceptions about health and the body shift accordingly [67]. Others noted the interplay between sociocultural features and pressure to improve one’s appearance, internalization of beauty ideals, appearance comparison tendencies, and their associations with body image concerns, which, in turn, were associated with disordered eating and exercise behaviors, with few sex differences [69]. Certainly, the interconnectedness between ethnic identity, societal pressures regarding thinness, internalization of societal beauty ideals, body image concerns, and disordered eating are not straightforward. For instance, ethnic identity was inversely related, and societal pressures regarding thinness were directly related to the internalization of societal beauty ideals; societal pressures regarding thinness were also related to greater body image concerns, and both internalization of societal beauty ideals and body image concerns were positively associated with disordered eating [70]. Norms differ by culture, where, for example, African American women were hesitant in their acceptance of dominant markers of health, aired an almost complete disregard for the thin ideal as a marker of “good” health and a positive body image, and were even suspicious of formal medical measurements of obesity [71]. The importance of our findings is that the current study assessed both genders across three countries characterized by very sparse body image literature, whereas the majority of studies that have examined such sociocultural variables and showed, via correlational, longitudinal, and experimental research, that they were linked to disordered eating [72], were amongst samples of predominantly European and American women.

### Strengths and Limitations of the Study

The key strengths of this study include that it links together a wide range of health status, healthy behavior, and sociodemographic characteristics to include weight satisfaction, body image concern, BMI, health awareness, and others. We assessed large and diverse samples in terms of academic study disciplines, year or level of study at university, country, ethnicity, religion, gender, and economic differentials. In addition, the findings were presented stratified for country and gender. Such characteristics maximize the validity and generalizability of the findings and allow a broad interpretation of body image concerns among university students.

Notwithstanding, this study has limitations. It is cross-sectional, so causality could not be inferred between sociodemographic, health status, and healthy behavior characteristics on the one hand, and the outcomes of weight satisfaction and body image concerns on the other. Due to respondent burden and the study being a general health survey administered at the end of lectures, some variables were measured by a single question, for example, weight satisfaction, whilst other variables that could be important in their relationships with body image concern and weight satisfaction beyond the influence of BMI (e.g., personality, behavioral and psychological distresses) were not examined, although we assessed depressive symptoms. In addition, BMI derived from weight and height was assessed differently: in Egypt, weight and height were measured, whereas, in Palestine and Finland, weight and height were self-reported. In terms of the method of administration of the questionnaire, although it is documented that electronic online survey administered over the internet result in lower response rates than paper surveys administered face to face, the lower response rate in Finland compared to the higher rates in Egypt and Palestine could still have influenced the findings. Likewise, the differences in the study populations, with regards to age and other specific characteristics of the participants and year of data collection, could have introduced some biases. Therefore, it might not be ideal to present the results together; however, our findings were presented stratified for countries, and the different model fits (max rescaled R-square in linear regression and adjusted R-square in logistic regression) did not differ between the three countries. To date, there is a knowledge gap in comparative body image research from countries with different levels of human development and from societies with different cultural conceptions [28]. The current study bridged this gap the first time. Further research is warranted.

## 5. Conclusions

Depressive symptoms and BMI were the most prominent predictors of body image concern when compared across Palestine, Egypt, and Finland. Country-specific consistent results were found using a body image concern score based on the body shape questionnaire. Healthy behaviors (healthy nutrition, physical activity, health awareness) were generally not associated with body image concern or weight satisfaction. The use of a single question on weight satisfaction might not generate consistent results due to culture-specific differences between the countries due to different interpretations. Further research is necessary to compare the use of body image questionnaires across different cultures and countries.

## Figures and Tables

**Table 1 nutrients-11-02860-t001:** Characteristics of the sample by country.

Total	Egypt		Palestine		Finland	
	2612		1219		1092	
Characteristic	*n*	%	*n*	%	*n*	%
Sociodemographic						
Sex (Female)	1355	51.90	717	58.82	770	70.51
Age group (years)						
16–18	1131	43.43	147	12.06	10	0.92
19–21	1369	52.57	755	61.94	578	52.93
22–30	104	3.99	317	26.00	504	46.15
Religiosity (Very important)	2476	95.08	1127	92.45	437	40.02
Income sufficiency (Always or mostly sufficient)	1993	76.54	747	61.28	455	41.67
Health status						
Self-rated general health (Excellent, very good and good)	2111	81.07	277	22.72	1012	92.67
Depressive symptoms						
85%-cutoff point *	371	14.25	243	19.93	122	11.17
Healthy Behavior						
Moderate PA (Achieved guidelines)	789	30.30	201	16.49	182	16.67
Vigorous PA (Achieved guidelines)	636	24.42	312	25.59	309	28.30
Healthy nutrition (Achieved guidelines)	173	6.64	147	12.06	278	25.46
Health awareness (Very much/some extent)	1925	73.92	951	78.01	932	85.35
Weight satisfaction and Body image concern *^a^*						
Weight satisfaction (Very satisfied or somewhat satisfied)	943	36.10	766	62.84	370	33.88
Body image concern *^a^*						
Mean, SD **	2.73	0.41	2.69	0.44	2.72	0.43
Cronbach’s Alpha	0.80		0.86		0.90	

*^a^* Mean and standard deviation; * M-BDI with cutoff point of >85% percentile of overall sample; ** Body image concern presented as numerical value (mean of the log transformed score and standard deviation); SD: standard deviation.

**Table 2 nutrients-11-02860-t002:** Weight satisfaction and body image concern by BMI category among university students in Egypt, Palestine, and Finland.

	Egypt			Palestine			Finland		
	BMI Category *								
	Below Normal	Normal	Overweight	Below Normal	Normal	Overweight	Below Normal	Normal	Overweight
*n* (%)	104 (4.1)	1711 (67.1)	737 (28.9)	68 (6.2)	756 (69.2)	268 (24.5)	39 (3.6)	845 (78.3)	195 (18.1)
Weight satisfaction **									
Not satisfied	42 (2.6)	**1196 (73.2)**	397 (24.3)	**34 (8.3)**	203 (49.8)	**171 (41.9)**	**27 (3.8)**	**619 (86.1)**	73 (10,2)
Satisfied	**62 (6.7)**	520 (56.3)	**342 (37.0)**	34 (5.0)	**553 (73.2)**	97 (14.2)	12 (3.3)	226 (62.8)	**122 (33.9)**
Body image concern ***									
Mean (SD)	2.61 (0.34)	2.62 (0.36)	2.99 (0.41)	2.51 (0.32)	2.61 (0.40)	2.97 (0.46)	2.51 (0.45)	2.68 (0.40)	2.92 (0.46)

* BMI categories were: below normal (≤18 kg/m^2^), normal (18–25 kg/m^2^), overweight (>25 kg/m^2^); measured BMI in Egypt, reported BMI in Palestine and Finland. Missing BMI values: Egypt *n* = 53, Palestine *n* = 127; Finland *n* = 13; ** Weight satisfaction derived from 4 categories; bolded cells indicate values that are above the sample’s average for the given country to visualize, for each BMI category in the three different countries, whether weight satisfaction or weight dissatisfaction were overrepresented; *** log-transformed score; Mean and standard deviation are presented (SD: standard deviation).

**Table 3 nutrients-11-02860-t003:** Multivariable logistic regression of health status, healthy behavior, and sociodemographic characteristics on weight satisfaction *****, stratified by country and sex.

	Egypt		Palestine		Finland	
	Men	Women	Men	Women	Men	Women
	OR (99.9% CI)	OR (99.9% CI)	OR (99.9% CI)	OR (99.9% CI)	OR (99.9% CI)	OR (99.9% CI)
Characteristic *n*	1249	1355	502	717	322	770
Health status						
Self-rated general health	0.64 (0.37; 1.13)	0.86 (0.47; 1.56)	0.56 (0.10; 3.21)	0.53 (0.11; 2.43)	2.79 (0.53; 14.72)	2.35 (0.58; 9.46)
Depressive symptoms	**0.98 (0.97; 0.99)**	**0.97 (0.96; 0.99)**	1.01 (0.99; 1.03)	1.02 (0.99; 1.03)	**0.96 (0.93; 0.99)**	**0.96 (0.94; 0.98)**
BMI	0.98 (0.93; 1.04)	**0.93 (0.88; 0.98)**	**1.08 (1.00; 1.15)**	**1.22 (1.10; 1.34)**	**0.78 (0.66; 0.94)**	**0.67 (0.58; 0.76)**
Healthy behavior						
Moderate PA	0.91 (0.56; 1.47)	1.26 (0.80; 1.98)	0.67 (0.25; 1.82)	0.83 (0.34; 2.02)	0.47 (0.10; 2.32)	0.79 (0.33; 1.87)
Vigorous PA	0.71 (0.43; 1.16)	0.93 (0.54; 1.59)	1.81 (0.79; 4.39)	1.15 (0.49; 2.68)	0.96 (0.30; 3.04)	0.68 (0.33; 1.42)
Healthy nutrition	1.52 (0.58; 3.99)	0.97 (0.46; 2.02)	0.66 (0.23; 1.85)	0.64 (0.25; 1.62)	1.10 (0.28; 4.30)	0.92 (0.46; 1.85)
Health awareness	1.07 (0.65; 1.75)	1.34 (0.85; 2.11)	0.49 (0.20; 1.21)	0.84 (0.41; 1.71)	0.71 (0.20; 2.47)	0.93 (0.36; 2.41)
Sociodemographic						
Age	0.93 (0.81; 1.07)	0.96 (0.81; 1.13)	0.98 (0.86; 1.12)	0.96 (0.87; 1.06)	1.02 (0.91; 1.15)	1.03 (0.97 1.10)
Income sufficiency	0.62 (0.38; 1.01)	0.74 (0.45; 1.21)	0.77 (0.36; 1.62)	1.37 (0.70; 2.64)	0.90 (0.33; 2.10)	1.02 (0.56; 1.88)
Religiosity	1.23 (0.55; 2.77)	1.25 (0.35; 4.45)	1.29 (0.42; 4.00)	0.59 (0.16; 2.23)	0.77 (0.28; 2.10)	0.92 (0.50; 1.70)
Max rescaled R-square	0.08	0.10	0.10	0.20	0.28	0.37

* Reference group is being satisfied; OR: odds ratio; CI: confidence interval; PA: physical activity; bolded cells indicate statistical significance (*p* < 0.001).

**Table 4 nutrients-11-02860-t004:** Multivariable linear regression of health status, healthy behavior, and sociodemographic characteristics on body image concern *****, stratified by country and sex.

	Egypt		Palestine		Finland	
	Men	Women	Men	Women	Men	Women
	*ß* (99.9% CI)	*ß* (99.9% CI)	*ß* (99.9% CI)	*ß* (99.9% CI)	*ß* (99.9% CI)	*ß* (99.9% CI)
Characteristic *n*	1249	1355	502	717	322	770
Health status						
Self-Rated general health	0.04 (−0.04; 0.12)	0.04 (−0.05; 0.13)	0.03 (−0.28; 0.35)	−0.03 (−0.28; 0.21)	−0.10 (−0.31; 0.12)	0.08 (−0.11; 0.26)
Depressive symptoms	**0.01 (0.01; 0.01)**	**0.01 (0.01; 0.01)**	**0.01 (0.00; 0.01)**	**0.01 (0.00; 0.01)**	**0.01 (0.01; 0.01)**	**0.01 (0.01; 0.01)**
BMI	**0.04 (0.03; 0.05)**	**0.05 (0.04; 0.06)**	**0.03 (0.02; 0.04)**	**0.04 (0.03; 0.05)**	**0.03 (0.02; 0.05)**	**0.04 (0.03; 0.06)**
Healthy behaviour						
Moderate PA	−0.03 (−0.04; 0.11)	−0.06 (−0.11; 0.02)	0.02 (−0.16; 0.19)	−0.03 (−0.18; 0.12)	0.09 (−0.07; 0.25)	0.01 (−0.11; 0.12)
Vigorous PA	0.01 (−0.07; 0.08)	0.01 (−0.08; 0.10)	−0.01 (−0.16; 0.13)	−0.06 (−0.21; 0.08)	−0.02 (−0.15; 0.11)	−0.02 (−0.12; 0.08)
Healthy Nutrition	−0.04 (−0.19; 0.11)	−0.00 (−0.11; 0.12)	−0.12 (−0.31; 0.01)	0.03 (−0.13; 0.19)	0.02 (−0.12; 0.17)	0.01 (−0.08; 0.11)
Health Awareness	−0.02 (−0.09; 0.06)	−0.04 (−0.11; 0.03)	−0.15 (−0.32; −0.02)	−0.02 (−0.15; 0.09)	0.04 (−0.10; 0.17)	0.07 (−0.06; 0.20)
Sociodemographic						
Age	−0.00 (−0.03; 0.02)	0.01 (−0.01; 0.04)	0.00 (−0.02; 0.02)	−0.00 (−0.02; 0.01)	−0.01 (−0.02; 0.01)	−0.00 (−0.01; 0.00)
Income sufficiency	0.05 (−0.03; 0.12)	0.02 (−0.06; 0.10)	−0.02 (−0.16; 0.11)	0.08 (−0.04; 0.19)	−0.03 (−0.14; 0.09)	−0.07 (−0.16; 0.02)
Religiosity	0.00 (−0.12; 0.13)	0.06 (−0.14; 0.27)	0.12 (−0.08; 0.32)	−0.08 (−0.31; 0.15)	−0.02 (−0.13; 0.08)	0.00 (−0.08; 0.09)
Adj. R-Square	0.23	0.32	0.20	0.26	0.32	0.35

* log-transformed scores; CI: confidence interval; PA: physical activity; bolded cells indicate statistical significance (*p* < 0.001).

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
