# Peer review of "Association of Health Status and Health Behaviors with Weight Satisfaction vs. Body Image Concern: Analysis of 5888 Undergraduates in Egypt, Palestine, and Finland"

_nutrients, 2019, doi:10.3390/nu11122860_

Round 1

Reviewer 1 Report

Although there are likely some interesting data collected in this study, I have some concerns about the manuscript.

First, the central dependent variable should be more clearly described and then measured in such a way that the authors can indicate that it is a valid form of measurement.  I am assuming it is body image concern?  (Although sometimes the one-item weight satisfaction scale was used to make predictions). 

In general, with the exception of two scales, the measures used were developed for the present study so their validity is questionable. 

There is insufficient theoretical rationale provided for comparing these particular countries using the variables that were measured. 

The response rate was so low for the group in Finland that it is not valid to compare the responses of that group with the other two where response rate was very high.  (The fact that it was so high is also an issue of concern as that is not typical.)

A more succinct, clearer presentation of the data is desirable.  It is not always clear what is represented on some of the tables.

Author Response

Reviewer 1

Although there are likely some interesting data collected in this study, I have some concerns about the manuscript.

First, the central dependent variable should be more clearly described and then measured in such a way that the authors can indicate that it is a valid form of measurement. I am assuming it is body image concern? (Although sometimes the one-item weight satisfaction scale was used to make predictions).

l We have now amended the manuscript. The study examined two central dependent variables, namely body image concern and weight satisfaction. The former was measured by 8 items of the body shape questionnaire [45, 46]; the latter by 1 item [44]. (Line 150-152)

Cooper, P.J.; Taylor, M.J.; Cooper, Z.; Fairburn, C.G. The development and validation of the Body Shape Questionnaire. International Journal of Eating Disorders. 1986, 6, 485-494. Anon. Body shape questionnaire (BSQ) and its shortened forms Psyctc, 2011 Available from https://www.psyctc.org/tools/bsq/ Accessed Dec 2017. El Ansari W, Labeeb S, Moseley L, Kotb S, El-Houfy A. Physical and Psychological Well-being of University Students: Survey of Eleven Faculties in Egypt. Int J Prev Med. 2013; 4(3): 293-310.

l In addition, we have now removed the variable entitled ‘Self-Perception of One’s Weight’ across the abstract, manuscript and Tables in order to make it absolutely clear to the reader that there are only two dependent variables, namely body image concern and weight satisfaction

In general, with the exception of two scales, the measures used were developed for the present study so their validity is questionable.

l None of the measures were developed for the present study. We have now amended the manuscript and added the appropriate references for all the measures used. (pp. 4-5)

There is insufficient theoretical rationale provided for comparing these particular countries using the variables that were measured.

l We have now amended the manuscript highlighting sufficient theoretical rationale provided for comparing these particular countries using the variables that were measured (bottom of p. 2 and top of p. 3)

The response rate was so low for the group in Finland that it is not valid to compare the responses of that group with the other two where response rate was very high. (The fact that it was so high is also an issue of concern as that is not typical.)

l We thank the reviewer. It is documented that electronic on line survey administered over the internet result in lower response rates than paper surveys that are administered face to face.

 l We have now added this point to the limitations section of the manuscript (p. 12)

A more succinct, clearer presentation of the data is desirable. It is not always clear what is represented on some of the tables.

l We have now amended the manuscript. We have also simplified the tables as we have now removed the variable entitled ‘Self-Perception of One’s Weight’ from the Tables and the manuscript

Other changes

l New references have been added

l All new references have been added to the reference list and formatted accordingly

Reviewer 2 Report

The article studies the relationships among individual health status; body image, physical activity and sociodemographics using a sample of university students in three countries. The article is well-written and well-researched. I have two minor comments. 1) the manuscript should include in the literature review the article by Demagistris et al. "The Impact of Body Image on the WTP Values for Reduced-Fat and Low-Salt Content Potato Chips among Obese and Non-Obese Consumers" in Nutrients. 2) The survey is carried out using different methods (on line survey in Finland and a self-administrated questionnaire in Egypt and Palestine). I am aware about the potential confusing effect between country and administration method of questionnaire. In my opinion this evidence should be mentioned in the limitations of the study.

Author Response

Ms. Ref. No.:  nutrients-571592

Title: Association of Health Status and Health Behaviors with Weight
Satisfaction and Body Image Concern: Analysis of 5888 Undergraduates in
Egypt, Palestine and Finland.

Dear Editor

We thank the reviewers for their constructive comments. Please find below our responses and amendments in bold to each of the reviewers’ comments.

Within the revised manuscript, our amendments are highlighted in red ink.

We hope that these are to the satisfaction of the reviewers and that the manuscript can now be accepted for publication.

Thank you

Reviewer 2

The article studies the relationships among individual health status; body image, physical activity and sociodemographics using a sample of university students in three countries. The article is well-written and well-researched.

l We agree with and thank the reviewer

I have two minor comments.

1) the manuscript should include in the literature review the article by Demagistris et al. "The Impact of Body Image on the WTP Values for Reduced-Fat and Low-Salt Content Potato Chips among Obese and Non-Obese Consumers" in Nutrients.

l We thank the reviewer, and have now added the article in the manuscript (line 40)

2) The survey is carried out using different methods (on line survey in Finland and a self-administrated questionnaire in Egypt and Palestine). I am aware about the potential confusing effect between country and administration method of questionnaire. In my opinion this evidence should be mentioned in the limitations of the study.

l We thank the reviewer. It is documented that electronic on line survey administered over the internet result in lower response rates than paper surveys administered face to face

 l We have now added this point to the limitations section of the manuscript (p. 12)

Other changes

 l New references have been added

l All new references have been added to the reference list and formatted accordingly

Reviewer 3 Report

This is a very well written paper. I have some remarks.

The abstract, although unstructured, should follow the classic steps of background, methods, results and conclusions.

The rationale of choosing the specific countries should be given more clearly justifying the purpose of the study. Also, the title should be probably amended, into: is it versus ? or maybe it should be amended.

There are many words given throughout the text for the outcomes. This report should focus on specific ones, for this article to be more easy to the reader.

Primary and secondary outcomes should be a priori set and than studied.

The difference in the study population concerning age, years of collection of participants, response rates, specific categories of participants, number and other characteristics insert biases. Authors should let the reader know how they dealt with this.

Direct comparison with similar studies is missing.

The conclusion should be more strictly adhere to the results.

Author Response

Ms. Ref. No.:  nutrients-571592

Title: Association of Health Status and Health Behaviors with Weight
Satisfaction and Body Image Concern: Analysis of 5888 Undergraduates in
Egypt, Palestine and Finland.

Dear Editor

We thank the reviewers for their constructive comments. Please find below our responses and amendments in bold to each of the reviewers’ comments.

Within the revised manuscript, our amendments are highlighted in red ink.

We hope that these are to the satisfaction of the reviewers and that the manuscript can now be accepted for publication.

Thank you

Reviewer 3

The abstract, although unstructured, should follow the classic steps of background, methods, results and conclusions.

l We have now amended the manuscript and structured the abstract. We added an introductory sentence and another sentence to outline the methods used. (p. 1)

The rationale of choosing the specific countries should be given more clearly justifying the purpose of the study.

l We have now amended the manuscript highlighting sufficient theoretical rationale provided for comparing these particular countries using the variables that were measured (Line 57-99)

Also, the title should be probably amended, into: is it versus ? or maybe it should be amended.

l We have now amended the title as suggested. (p. 1)

There are many words given throughout the text for the outcomes. This report should focus on specific ones, for this article to be more easy to the reader.

Primary and secondary outcomes should be a priori set and than studied.

l We thank the reviewer. We have now amended the manuscript. The study examined two central dependent variables, namely body image concern and weight satisfaction.

 l In addition, we have now removed the variable entitled ‘Self-Perception of One’s Weight’ across the abstract, manuscript and Tables in order to make it absolutely clear to the reader that there are only two dependent variables, namely body image concern and weight satisfaction

The difference in the study population concerning age, years of collection of participants, response rates, specific categories of participants, number and other characteristics insert biases. Authors should let the reader know how they dealt with this.

l We have now amended the manuscript and added this point to the limitations section of the manuscript (p. 12). To date, there is an evident gap in comparisons of body image across countries of different levels of development and from societies with different cultural conceptions. The current paper bridges this gap for the first time. Further research is warranted.

Direct comparison with similar studies is missing.

l We thank the reviewer. We searched the literature once again to find similar articles. We only found one article published in 2019 comparing body image and eating distress in Japan and Finland (ref 28). We have now cited this article. Unfortunately, the authors of this article presented their findings differently and the results cannot be directly compared to ours.

The conclusion should be more strictly adhere to the results.

l We have now amended the manuscript

Other changes

 l New references have been added

l All new references have been added to the reference list and formatted accordingly

Round 2

Reviewer 1 Report

The authors addressed some of my original concerns in this revised manuscript.  However, there still needs to be further improvement.

The central question of the study is not clearly stated in the abstract.

Although the authors do a better job discussing the importance of culture, it would be a stronger study if they had more specific hypotheses. 

How was the study advertised to participants?

Why was the scoring of weight satisfaction collapsed from four responses to two?  More variability would make it easier to find significant predictors.

On Table 1 the results for weight satisfaction are unclear.

The logic of dividing people by BMI category and then assessing their weight satisfaction should be provided.  Further, the results in Table 2 are confusing.  It is unclear why the values are compared to the sample’s average and then represented in bold.

There are grammatical issues throughout the paper that should be addressed.

Author Response

Ms. Ref. No.:  nutrients-571592, second round

Title: Association of Health Status and Health Behaviors with Weight
Satisfaction and Body Image Concern: Analysis of 5888 Undergraduates in
Egypt, Palestine and Finland.

Dear Editor

We thank the reviewers for their constructive comments. Please find below our responses and amendments in bold to each of the reviewers’ comments.

Within the revised manuscript, our amendments are highlighted in red ink.

We hope that these are to the satisfaction of the reviewers and that the manuscript can now be accepted for publication.

Thank you

 Reviewer 1

The authors addressed some of my original concerns in this revised manuscript. 

l We thank the reviewer.

However, there still needs to be further improvement.

The central question of the study is not clearly stated in the abstract.

l We have now amended the abstract (Lines 17-18), and the manuscript. (Line 94)

Although the authors do a better job discussing the importance of culture, it would be a stronger study if they had more specific hypotheses. 

l The current study is an explorative study. As such, no specific hypothesis was formulated.

Why was the scoring of weight satisfaction collapsed from four responses to two?  More variability would make it easier to find significant predictors.

l We thank and agree with the reviewer. However, this variable was collapsed because of the small numbers in the two most extreme groups (very satisfied and very dissatisfied).

l In addition, separate presentations of these two extreme values would not have added significant information due to their small numbers.

On Table 1 the results for weight satisfaction are unclear.

l The results for weight satisfaction in Table 1 denote the numbers and percentages of participants who felt very satisfied or somewhat satisfied with their current weight. (Table 1)

The logic of dividing people by BMI category and then assessing their weight satisfaction should be provided.  Further, the results in Table 2 are confusing.  It is unclear why the values are compared to the sample’s average and then represented in bold.

l The first objective of the study was to assess weight satisfaction and body image concern by BMI category. (Lines 95 and 100)

l In addition, we have now also added the sentenceUnsatisfied students were overrepresented in the below normal BMI category (≤18 kg/m2) in Egypt but not in Palestine and Finland.’ (Lines 241-242)

l We have now also added the sentencebolded cells indicate values that are above the sample’s average for the given country to visualize, for each BMI category in the three different countries, whether weight satisfaction or weight dissatisfaction were overrepresented’ (Lines 247-248)

There are grammatical issues throughout the paper that should be addressed.

l We have now amended the manuscript.

Reviewer 3 Report

I think that suggestions have been addressed successfully. 

Author Response

Ms. Ref. No.:  nutrients-571592, second round

Title: Association of Health Status and Health Behaviors with Weight
Satisfaction and Body Image Concern: Analysis of 5888 Undergraduates in
Egypt, Palestine and Finland.

Dear Editor

We thank the reviewers for their constructive comments. Please find below our responses and amendments in bold to each of the reviewers’ comments.

Within the revised manuscript, our amendments are highlighted in red ink.

We hope that these are to the satisfaction of the reviewers and that the manuscript can now be accepted for publication.

Thank you

 Reviewer 3

I think that suggestions have been addressed successfully. 

l We agree with and thank the reviewer.
